# Autophagy and Programmed Cell Death Modalities Interplay in HIV Pathogenesis

**DOI:** 10.3390/cells14050351

**Published:** 2025-02-28

**Authors:** Harpreet Kaur Lamsira, Andrea Sabatini, Serena Ciolfi, Fabiola Ciccosanti, Alessandra Sacchi, Mauro Piacentini, Roberta Nardacci

**Affiliations:** 1Departmental Faculty of Medicine, Saint Camillus International University of Health Sciences, 00131 Rome, Italy; harpreetkaur.lamsira@unicamillus.org; 2Department of Science, University ‘Roma Tre’, 00146 Rome, Italyserena.ciolfi@uniroma3.it (S.C.); alessandra.sacchi@uniroma3.it (A.S.); 3Department of Epidemiology, Preclinical Research and Advanced Diagnostics, National Institute for Infectious Diseases IRCCS ‘L. Spallanzani’, 00149 Rome, Italy; fabiola.ciccosanti@inmi.it (F.C.);; 4Department of Biology, University ‘Tor Vergata’, 00133 Rome, Italy

**Keywords:** autophagy, xenophagy, apoptosis, necroptosis, ferroptosis, pyroptosis, HIV

## Abstract

Human immunodeficiency virus (HIV) infection continues to be a major global health challenge, affecting 38.4 million according to the Joint United Nations Program on HIV/AIDS (UNAIDS) at the end of 2021 with 1.5 million new infections. New HIV infections increased during the 2 years after the COVID-19 pandemic. Understanding the intricate cellular processes underlying HIV pathogenesis is crucial for developing effective therapeutic strategies. Among these processes, autophagy and programmed cell death modalities, including apoptosis, necroptosis, pyroptosis, and ferroptosis, play pivotal roles in the host–virus interaction dynamics. Autophagy, a highly conserved cellular mechanism, acts as a double-edged sword in HIV infection, influencing viral replication, immune response modulation, and the fate of infected cells. Conversely, apoptosis, a programmed cell death mechanism, is a critical defense mechanism against viral spread and contributes to the depletion of CD4+ T cells, a hallmark of HIV/AIDS progression. This review aims to dissect the complex interplay between autophagy and these programmed cell death modalities in HIV-induced pathogenesis. It highlights the molecular mechanisms involved, their roles in viral persistence and immune dysfunction, and the challenges posed by the viral reservoir and drug resistance, which continue to impede effective management of HIV pathology. Targeting these pathways holds promise for novel therapeutic strategies to mitigate immune depletion and chronic inflammation, ultimately improving outcomes for individuals living with HIV.

## 1. Introduction

Human immunodeficiency virus-1 (HIV-1) infection remains a significant global health burden, with approximately 38.4 million people living with the virus worldwide [1]. The HIV-1 genome is a single-stranded RNA virus belonging to the Retroviridae family that primarily infects CD4+ T cells, leading to progressive immune system deterioration and ultimately causing acquired immunodeficiency syndrome (AIDS) [2]. After viral entry into host cells, the viral RNA is reverse-transcribed into DNA by the enzyme reverse transcriptase and integrated into the host genome by the action of integrase. For its replication, HIV-1 uses the host’s cellular machinery. HIV-1 pathogenesis is characterized by a high rate of viral replication, immune evasion, and the gradual depletion of CD4+ T cells, resulting in a weakened immune response and increased susceptibility to opportunistic infections and malignancies [3].

The HIV-1 genome is approximately 9.2 kb in length and contains three structural genes essential for the replication of HIV-1 (gag, pol, and env), and several overlapping Open Reading Frames (ORFs) that encode regulatory proteins (Tat, Rev, and Nef) and accessory proteins (Vif, Vpr, Vpu, and Vpx), which are crucial for the virus’s life cycle. Several viral proteins play crucial roles in its ability to evade the host immune response. Vpr (Viral Protein R) is involved in the nuclear import of the viral genome, modulation of cell cycle arrest, and induction of apoptosis in infected cells, aiding in viral persistence and pathogenesis [4]. Vpu (Viral Protein U) counteracts host cellular defenses by downregulating CD4 receptors and enhancing the release of newly formed virions, promoting viral spread [5]. Env is a glycoprotein critical for viral entry into host cells, mediating the attachment and fusion of the virus with the target cell membrane. Tat (Trans-Activator of Transcription) is essential for enhancing viral transcription by recruiting host transcription factors to the viral promoter, thereby increasing viral RNA synthesis [6]. Additionally, the Nef (Negative Factor) protein manipulates host cellular pathways by downregulating surface receptors such as CD4 and MHC-I, facilitating immune evasion and enhancing viral infectivity [5].

Together, these proteins form a complex network that drives HIV-1 pathogenesis, viral persistence, and the failure of the host immune response, underscoring the challenges in developing effective therapies and vaccines against HIV-1.

Despite advancements in antiretroviral therapy (ART), which have significantly improved the quality of life and longevity of HIV-1-infected individuals, challenges such as viral persistence, drug resistance, and long-term side effects persist [7,8,9]. A deeper understanding of the molecular mechanisms governing HIV-1 pathogenesis is essential for developing novel therapeutic interventions to eradicate the virus from its reservoirs or to prevent infection.

In this context, the processes of autophagy and apoptosis have emerged as central players in the host–virus interaction dynamics, exerting profound effects on viral replication, immune response modulation, and disease progression [10,11].

## 2. Autophagy in HIV-1 Pathogenesis

Autophagy is a fundamental cellular process for degrading and recycling intracellular components. It is an essential catabolic process, where intracellular components are delivered to lysosomes for degradation, thus ensuring cellular homeostasis by degrading damaged organelles and proteins and replenishing cellular energy stores during nutrient scarcity [12,13]. A complex interplay exists between the virus and the host’s autophagic machinery [14]. HIV-1 both exploits and inhibits autophagy to facilitate its life cycle, posing challenges and opportunities for therapeutic intervention. In particular, autophagy was shown to be effective in controlling HIV-1 infection among long-term nonprogressors (LTNPs) and elite controllers (ECs). These individuals maintain stable health without antiretroviral therapy. It was found that peripheral blood mononuclear cells (PBMCs) from LTNPs and ECs exhibited higher levels of autophagic activity compared to normal progressors [15]. Recent studies have demonstrated that patients under ART display excessive use of glycolysis, whereas ECs are dependent on mitochondrial input [16].

Autophagy involves several steps: initiation, nucleation, elongation, and fusion with lysosomes. Key proteins, including those from the ATG family, regulate these steps [17,18,19]. The process begins with forming a phagophore, by the activity of the ULK complex, comprising ULK1, ULK2, ATG13, ATG101, and RB1CC1. The PIK3C3 complex, including BECN1, ATG14, PIK3C3, and PIK3R4, assists in phagophore elongation [20,21,22,23]. The ULK complex function is finely modulated: for instance, it is inhibited by the mechanistic target of rapamycin complex 1 or mammalian target of rapamycin complex 1 (MTORC1, named mTORC1)-mediated phosphorylation of ULK1, ULK2, and ATG13. The activity of the BECN1-PIK3C3 complex could be regulated by its interaction with several cofactors, such as UVRAG (UV radiation resistance-associated), ATG14, and AMBRA1 (autophagy and BECN1 regulator 1) [24].

Autophagosome formation and elongation involve additional factors like WIPI proteins, ATG2, ATG9, and a ubiquitin-like conjugation system involving ATG12, ATG5, and ATG16L1. Moreover, selective autophagy, targeting specific cargos, involves factors like SQSTM1/p62, NBR1, and OPTN [25,26]. Once generated, the autophagosome’s outer membrane fuses with a lysosome, enabling the degradation of content by lysosomal enzymes.

Several HIV-1 proteins, including Env, Tat, Vpr, Vif, Vpu, and Nef, play distinct roles in modulating autophagic processes to support viral replication and persistence while evading immune responses (Table 1).

**Table 1 cells-14-00351-t001:** The role of HIV proteins in the autophagy mechanism.

Protein	Role in the Autophagic Mechanism
Env (Envelope)	The Env protein is critical for viral entry into host cells and is involved in evading the host immune response. It disrupts autophagy by inhibiting autophagosome formation, reducing the degradation of viral particles by the host. This enables HIV to persist within cells by minimizing its exposure to immune detection mechanisms [27].
Tat (Trans-Activator of Transcription)	The Tat protein is essential for HIV replication, as it enhances transcription from the viral promoter. Tat also modulates autophagy in complex ways: it can induce autophagy in some contexts but often interferes with its completion (autophagic flux) to prevent the degradation of viral proteins. By manipulating autophagy, Tat facilitates an environment conducive to viral replication and persistence [28].
Vpr (Viral Protein R)	Induction of autophagy: Vpr has been shown to induce autophagy in infected cells. This process may contribute to the degradation of host cell components, providing nutrients for viral replication. Manipulation of autophagy for immune evasion: By inducing autophagy, Vpr may help the virus evade immune detection and contribute to the persistence of the infection [29,30].
Vif (Viral Infectivity Factor)	The Vif protein is responsible for counteracting host restriction factors (e.g., APOBEC3G) that would otherwise inhibit viral replication. Vif impairs autophagy by interfering with cellular pathways that would normally target the virus for degradation. This contributes to HIV’s ability to maintain infectivity and spread within host cells [31].
Vpu (Viral Protein U)	Modulation of autophagy: Vpu has been implicated in modulating autophagy, though its effects are complex and context dependent. Vpu may inhibit autophagy to prevent the degradation of viral particles and enhance virion release [32].
Nef	Inhibits autophagy to prevent viral degradation by interacting with BECN1, disrupts antigen processing and presentation via MHC molecules, downregulates CD4 and MHC-I molecules, prevents superinfection, facilitates viral release, evades cytotoxic T cells, and enhances infectivity of new virus particles [33,34].

HIV-1 can evade autophagy via Nef, which promotes the ubiquitination of the autophagy inhibitor BCL2 through the E3 ubiquitin ligase Parkin (PRKN). This ubiquitination occurs in the endoplasmic reticulum (ER) and mitochondria but has different outcomes based on the location: ER-associated ubiquitinated BCL2 inhibits autophagy, helping HIV-1 to evade this defense mechanism. The Env protein, primarily responsible for viral entry, has been shown to induce autophagy in infected cells [27].

In fact, Env has been shown to induce autophagic activity in certain immune cells, such as macrophages and CD4+ T cells [35]. The interaction between gp120 and the CD4 receptor activates signaling pathways that promote autophagy [36].

However, HIV-1 also inhibits autophagosome maturation, thereby preventing the degradation of viral components and promoting viral survival. Tat, another key regulatory protein, can both activate and inhibit autophagy depending on the cellular context. By enhancing autophagy early in infection, Tat promotes viral replication, while later, it inhibits the autophagic flux to prevent viral degradation and immune recognition [28,37].

Early in HIV-1 infection, Tat enhances autophagy in host cells through the inhibition of the mechanistic target of rapamycin [mTOR], a master regulator that normally suppresses autophagy. For instance, a study published in *The Journal of Neuroscience* reported that HIV-1 Tat alters neuronal autophagy by modulating the mTOR pathway, leading to neurodegeneration [38]. Tat also activates AMP-activated protein kinase [AMPK], a cellular energy sensor that triggers autophagy when intracellular ATP levels are low. Activation of AMPK by Tat not only inhibits mTOR but also promotes autophagy directly by phosphorylating autophagy-related proteins. This dual activation ensures a robust induction of autophagic activity, which increases the availability of metabolites necessary for efficient viral replication [39]. Vpr exerts a multifaceted influence on autophagy. It can induce autophagy by targeting host cell pathways such as the DNA damage response, but it also contributes to mitochondrial dysfunction, leading to apoptosis [29,30]. Vif, known for its role in counteracting host restriction factors like APOBEC3G [40], also interacts with the autophagic machinery to protect HIV-1-infected cells from degradation. Vif suppresses autophagy by inhibiting autophagy, ensuring the survival of infected cells [31]. Vpu, primarily involved in downregulating CD4 and tetherin [BST-2] to enhance viral release [32]. The viral protein Nef has a pivotal role in enhancing the infectivity of new virus particles within the producer cell [33,34]; moreover, it downregulates CD4 and MHC-I molecules [41], preventing superinfection and facilitating viral release while evading cytotoxic T cell recognition. Nef is also involved in autophagy inhibition interacting with BECN1 to inhibit autophagosome formation [42].

On the other hand, autophagy can impact HIV-1 replication by degrading viral components. In particular, PINK1-mediated mitophagy may contribute to the autophagic HIV-1 replication restriction. PINK1 plays an important role in the mitophagy process. It is a mitochondrial serine/threonine protein kinase crucial for maintaining mitochondrial health. It has gained prominence for its role in mitochondrial quality control mechanisms, such as autophagy [specifically mitophagy] and apoptosis [43]. PINK1 is a 63 kDa precursor protein imported into mitochondria, where it is cleaved to form a 52 kDa mature protein. Under mitochondrial stress, the import and processing of PINK1 are inhibited, leading to its accumulation on the outer mitochondrial membrane. This accumulation activates downstream processes aimed at mitochondrial quality control [43,44]. It accumulates on damaged mitochondria, starting mitophagy, and recruits Parkin, leading to the ubiquitination of mitochondrial proteins; this marks the mitochondria for degradation through autophagy. PINK1-mediated mitophagy may contribute to the autophagic HIV-1 replication restriction by maintaining mitochondrial function and reducing oxidative stress, which is conducive to viral replication. Indeed, efficient mitophagy ensures the removal of damaged mitochondria, preventing the release of mitochondrial DNA and other damage-associated molecular patterns [DAMPs] that can trigger inflammation and support viral replication [45]. In addition, in other contexts such as Parkinson’s disease, PINK1-regulated mitophagy can influence the presentation of viral antigens and the activation of adaptive immune cell response [46].

Traditionally considered a cytoprotective mechanism, autophagy has also emerged as a modulator of the immune response, including inflammation. In the context of HIV-1 infection, autophagy plays a complex role, particularly through its interaction with the Stimulator of Interferon Genes [STING] pathway. STING is a key adaptor protein in the innate immune response to cytosolic DNA and mediates the production of type I interferons [IFNs] and pro-inflammatory cytokines. The STING pathway is primarily activated by the cyclic GMP-AMP synthase [cGAS] [47], which detects cytosolic double-stranded DNA [dsDNA]. In HIV-1 patients, STING activation is triggered by the presence of viral DNA in the cytoplasm, produced during reverse transcription. Upon binding to cyclic GMP-AMP [cGAMP], a second messenger produced by cGAS, STING translocates from the endoplasmic reticulum [ER] to the Golgi apparatus [48]. This translocation triggers downstream signaling, including the activation of TBK1 and IRF3, which drive the expression of type I IFNs and inflammatory cytokines [49].

Autophagy has a dual role in the regulation of STING-mediated inflammation in HIV-1 patients. On one hand, autophagy can degrade cytosolic DNA or STING-containing vesicles, reducing excessive inflammation. On the other hand, autophagy may exacerbate inflammation through STING activation under certain conditions: during HIV-1 infection, reverse transcription generates cytosolic viral DNA that can activate the cGAS-STING pathway. Autophagy contributes to this process by releasing mitochondrial DNA [mtDNA] into the cytoplasm as a result of mitochondrial dysfunction. This mtDNA further amplifies cGAS-STING activation, leading to an enhanced inflammatory response [50]. Autophagosomes act as carriers for STING trafficking to the Golgi apparatus in HIV-1-infected cells. This trafficking facilitates the amplification of the STING signaling cascade, resulting in elevated production of IFNs and pro-inflammatory cytokines. In HIV-1 patients, this can contribute to systemic inflammation and immune dysregulation [51]. HIV-1 infection alters the expression of autophagy-related proteins such as Beclin-1 and ATG5. These proteins play a role in promoting STING activation during DNA damage and infection. For example, Beclin-1 interacts with STING, enhancing its activity and promoting inflammation [52].

## 3. Cell Death Modalities in HIV Pathogenesis

Apoptosis or programmed cell death: As previously mentioned, HIV-1-induced changes in autophagy impact cellular homeostasis, leading to the accumulation of damaged organelles and proteins and ultimately contributing to cell dysfunction and apoptosis [53,54] This is particularly evident in CD4+ T cells, where impaired autophagy correlates with increased cell death and depletion, marking the progression to AIDS [49,50]. CD4+ T cells are also strongly depleted by the action of syncytia that are multinucleated cells formed by the fusion of HIV-1-infected cells with uninfected ones, contributing to the depletion of CD4+ T cells in HIV-1-infected individuals through apoptotic pathways [55,56]. It was also observed that the syncytial apoptosis is mediated through the mitochondrial pathway, as evidenced by the release of cytochrome c from mitochondria into the cytosol and the activation of caspase-3, a key enzyme in the execution phase of apoptosis. PML (Promyelocytic Leukemia Protein)—a key component of PML nuclear bodies (PML-NBs), which play a role in cellular processes such as DNA damage response and antiviral defense mechanisms [57]—is essential for the activation of the DNA damage response [DDR] in these syncytia. Specifically, PML facilitates the activation of ataxia–telangiectasia mutated [ATM] kinase, which in turn activates the tumor suppressor protein p53. Activated p53 leads to the transcription of pro-apoptotic genes, culminating in cell death. Knockdown of PML was shown to inhibit this ATM-dependent DDR, thereby suppressing apoptosis in HIV-1-induced syncytia [57].

Apoptosis serves as a fundamental defense mechanism against viral infections by eliminating infected cells and preventing viral spread. In the context of HIV-1 infection, apoptosis plays a dual role, contributing to both host defense and viral pathogenesis [58]. CD4+ T cells are the main apoptotic target thus favoring the pathology’s progression; however, apoptosis also limits viral replication by eliminating infected cells and restricting viral spread within the host. Apoptosis in CD4+ T cells during HIV-1 infection can be induced through the action of several viral proteins like Vpr, which disrupts mitochondrial membrane potential, leading to the activation of caspases and cell death [59]. Vif also has an indirect role in inducing apoptosis in uninfected bystander CD4+ T cells through the release of viral proteins and inflammatory cytokines, contributing to overall CD4+ T cell depletion [60]

Moreover, Tat, Vpr, and Nef can directly induce apoptosis or dysregulate apoptotic signaling pathways, thereby promoting viral persistence and pathogenesis [61] (Table 2) (Figure 1), while Env acts by inducing protein encoded by the HIV-1 genome [62].

**Table 2 cells-14-00351-t002:** The role of HIV proteins in the apoptotic mechanism.

Protein	Role in the Apoptotic Mechanism
Vpr	Direct induction of apoptosis by disrupting mitochondrial membrane potential, activating caspases, and causing cell death [61].
Tat	Direct induction of apoptosis or dysregulation of apoptotic signaling pathways, promoting viral persistence and pathogenesis [61].
Nef	Direct induction of apoptosis or dysregulation of apoptotic signaling pathways, promoting viral persistence [61].
Vpu	Initiates activation of several apoptotic pathways (TRAIL, TNF, Fas/FasL) via HIV exosomes or as soluble proteins, leading to cell death [63].
Env	Main apoptosis-inducing protein encoded by the HIV-1 genome, triggers proapoptotic signaling by acting on chemokine receptors [62].

**Figure 1 cells-14-00351-f001:**
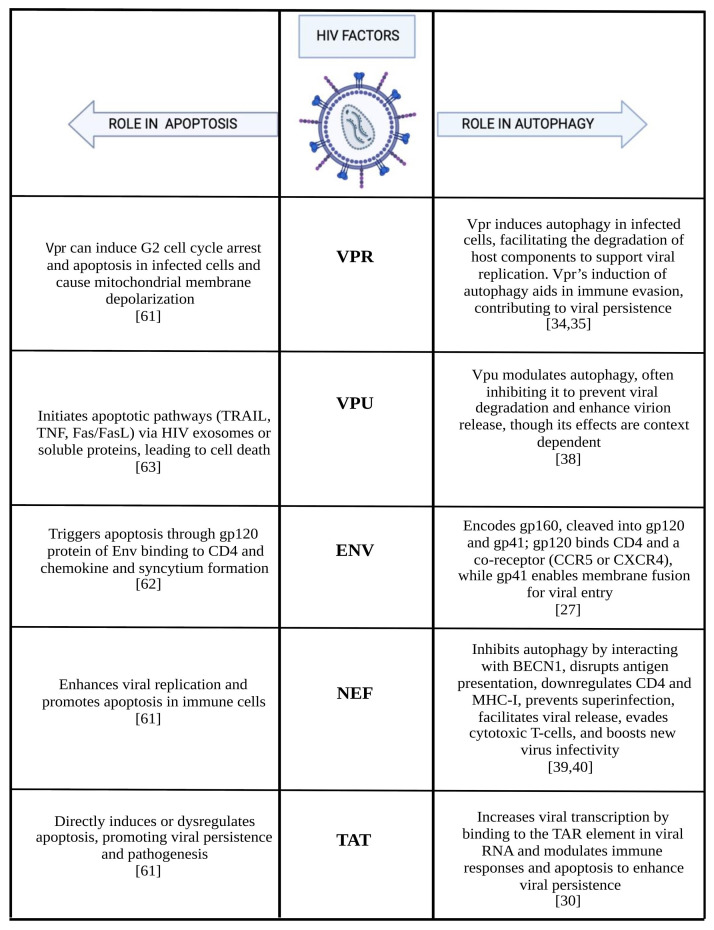
Role of HIV factors in the autophagy and in the apoptosis mechanism [27,30,34,35,38,39,40,61,62,63].

A higher rate of bystander death of non-permissive or resting CD4+ T cells takes place in the lymph nodes. Typically, retroviruses merge with target cells either as free-floating virions or via direct cell-to-cell transmission. This direct cell-to-cell transmission happens more frequently in the lymph nodes than in the bloodstream, thereby enhancing the virus’s infectivity [63]. The mechanism underlying the reduction in uninfected cells and influencing HIV-1’s survival is quite interesting since most retroviruses typically infect cells without killing their host [64]. It is likely that the killing of uninfected bystander CD4+ T cells often results in failed infections, thereby preventing HIV-1 replication and dissemination. The virus utilizes various mechanisms to accomplish this, including the activation of TNF-a, TRAIL, and Fas/FasL pathways through different viral proteins such as gp120, Tat, Vpu, and Vpr [64]. These proteins are usually released via HIV-1 exosomes or directly across the cell membrane as soluble proteins. The release and attachment of these viral proteins to uninfected cells initiate the activation of several apoptotic pathways, leading to cell death [65].

The previously mentioned PINK1 protein influences apoptosis in the context of HIV-1 infection through several mechanisms [66]: PINK1 helps in maintaining mitochondrial integrity, protecting cells from apoptosis. In HIV-1-infected cells, mitochondrial dysfunction and oxidative stress are prevalent [67]. PINK1’s role in preserving mitochondrial function can prevent the activation of apoptotic pathways, thereby reducing cell death [68]. It can also regulate the level and activity of pro-apoptotic proteins such as Bax and Bak. By maintaining mitochondrial membrane potential, PINK1 prevents the activation of these proteins, inhibiting the release of cytochrome c and subsequent caspase activation, which are key events in apoptosis [69]. PINK1’s protective role against apoptosis could theoretically mitigate CD4+ T cell loss [70], although this aspect requires further research.

Necroptosis is a form of programmed necrotic cell death driven by RIPK1 (Receptor-Interacting Protein Kinase 1), RIPK3 (Receptor-Interacting Protein Kinase 3), and MLKL 1 (Mixed Lineage Kinase Domain-Like Protein). The first is a crucial protein; upon activation by death receptors like TNFR1, RIPK1 can interact with RIPK3 to form a complex known as the necrosome, initiating necroptosis. In HIV-infected cells, the viral envelope glycoprotein gp41 can induce necroptosis through a pathway involving RIPK1, contributing to the depletion of CD4+ T cells. RIPK3 is essential for the execution of necroptosis. It interacts with RIPK1 to form the necrosome, leading to the phosphorylation and activation of MLKL. In the context of HIV, RIPK3-mediated necroptosis has been implicated in the death of infected cells, as well as uninfected bystander CD4+ T cells, thereby contributing to immune system decline. MLKL is the downstream effector in the necroptosis pathway. Once phosphorylated by RIPK3, MLKL translocates to the plasma membrane, causing its permeabilization and leading to cell death. In HIV infection, the activation of MLKL by RIPK3 is a critical step in the necroptosis of CD4+ T cells, contributing to the immunodeficiency characteristic of AIDS [71].

This pathway is activated in response to extracellular signals such as tumor necrosis factor-α [TNF-α] and contributes to inflammation by releasing damage-associated molecular patterns [DAMPs] [71] (Table 3).

In the context of HIV-1 infection, necroptosis contributes to immune activation and tissue damage. In Jurkat cells lacking Fas-associated protein with death domain [FADD], a key adaptor in apoptotic signaling, there was a significant increase in necroptosis upon HIV-1 infection. This indicates that in the absence of apoptosis, necroptosis serves as an alternative cell death mechanism [71].

Ferroptosis is characterized by the accumulation of lipid peroxides due to iron-catalyzed oxidative reactions. The key regulators of ferroptosis include glutathione peroxidase 4 [GPX4], which detoxifies lipid peroxides, and system Xc-, a cystine/glutamate antiporter that maintains intracellular glutathione levels. Dysregulation of these pathways leads to ferroptotic cell death [72]. HIV-1 infection induces metabolic reprogramming and oxidative stress, creating conditions favorable for ferroptosis. Elevated levels of iron and ROS, along with impaired antioxidant defenses, have been observed in HIV-1-infected cells [73]. Ferroptosis contributes to the loss of immune cells and exacerbates tissue damage, particularly in lymphoid and mucosal tissues critical for immune function [74] (Table 3).

Pyroptosis is driven by inflammasome activation, leading to the cleavage of gasdermin D (GSDMD) by caspase-1. Activated GSDMD forms pores in the plasma membrane, causing cell lysis and the release of pro-inflammatory cytokines such as IL-1β and IL-18. The majority of CD4 T cells die not from productive HIV-1 infection but from abortive infection, where the virus initiates replication but fails to complete it. This phenomenon was observed also in simian immunodeficiency virus (SIV)-infected rhesus macaques. SIV infection triggers the activation of caspase-1, an enzyme central to the pyroptotic pathway, resulting in the cleavage of gasdermin D and the subsequent formation of membrane pores that lead to cell lysis [75]. This incomplete replication leads to the accumulation of viral DNA, which is sensed by the host cell, triggering activation of caspase-1 within inflammasomes. Moreover, direct cell-to-cell transmission of HIV-1 is essential to induce pyroptosis in CD4 T cells derived from lymphoid tissues. In contrast, exposure to cell-free HIV-1 particles, even in large quantities, does not trigger pyroptosis [76] [Table 3]. This activation induces pyroptosis, resulting in cell death and the release of pro-inflammatory cytokines, notably interleukin-1β (IL-1β). The release of these inflammatory signals attracts more CD4 T cells to the site, perpetuating a cycle of infection and cell death. This discovery links the two hallmark events of HIV-1 infection: CD4 T cell depletion and chronic inflammation. By identifying pyroptosis as a key mechanism, the study suggests that targeting components of this pathway, such as caspase-1, could offer new therapeutic strategies to preserve CD4 T cells and mitigate inflammation in HIV-1-infected individuals [76]. This abortive infection of CD4+ T cells inducing pyroptosis contributes to chronic inflammation and immune cell depletion. The release of pro-inflammatory cytokines and DAMPs during pyroptosis amplifies immune activation. This creates a vicious cycle of inflammation and cell death, driving HIV-1 pathogenesis and progressive immune dysfunction [77].

**Table 3 cells-14-00351-t003:** Cell death modalities and major roles in HIV pathogenesis.

Cell Death Modalities	Major Role in HIV Pathogenesis
Apoptosis	Induced by HIV proteins like Tat and Vpr, leading to CD4+ T cell depletion through mitochondrial disruption, caspase activation, and DNA damage. This contributes to immune system collapse and chronic inflammation [61].
Pyroptosis	Predominantly occurs in abortively infected CD4+ T cells. Driven by inflammasome activation and caspase-1, it releases pro-inflammatory cytokines (IL-1β, IL-18), amplifying immune activation and chronic inflammation while contributing to T cell depletion [76].
Necroptosis	Promoted by RIPK1 and RIPK3 activation, resulting in the release of DAMPs like HMGB1 and ATP. These amplify immune activation, disrupt tissue homeostasis, and exacerbate systemic inflammation and microbial translocation in HIV-infected individuals [71].
Ferroptosis	Driven by iron-catalyzed lipid peroxidation and oxidative stress. HIV-induced dysregulation of autophagy enhances ferritinophagy, increasing free iron and lipid peroxidation, which exacerbate immune cell death and tissue damage in mucosal and lymphoid tissues [72,73,74].

## 4. Interplay Between Autophagy and Cell Death Mechanisms

Autophagy and apoptosis demonstrate a complex interaction in HIV-1 infection. Autophagy can either support cell survival by clearing damaged organelles or promote apoptosis under prolonged stress [78,79]. Proteins such as BECN1 are involved in both processes, linking their regulation. HIV-1 proteins like Nef and Tat manipulate BECN1 to modulate autophagy and apoptosis, enhancing viral replication and immune cell death [80,81,82]. Similarly, Bcl-2 family proteins, crucial in apoptosis, are dysregulated by HIV-1-induced oxidative stress, tipping the balance toward apoptosis [83,84].

P53, a tumor suppressor protein, influences both pathways: it activates autophagy-related genes but can also inhibit autophagy or induce apoptosis in response to stress [85]. Tat enhances p53-mediated apoptosis, contributing to CD4+ T cell depletion [86]. Caspases, central to apoptosis, cleave autophagy proteins like BECN1, further inhibiting autophagy and promoting cell death [87].

HIV-1 modulates pathways to support viral survival. Tat activates mTOR, inhibiting autophagy and promoting viral persistence [88]. Vpr induces apoptosis via mitochondrial damage, Bcl-2 family protein dysregulation, and DNA damage, particularly in CD4+ T cells [89,90,91]. It also activates DNA damage response kinases, amplifying apoptosis signals. Conversely, Nef prolongs infected cell survival by activating the anti-apoptotic PI3K/Akt pathway and promoting anti-apoptotic proteins like Bcl-2. Simultaneously, Nef induces apoptosis in bystander cells by upregulating Fas ligand and pro-inflammatory cytokines [92,93]. HIV-1-induced ubiquitination of mitochondrial proteins like BCL2 influences apoptosis and mitochondrial function. For example, BCL2 ubiquitination at mitochondria promotes apoptosis [94,95,96].

HIV-1 exploits ubiquitination to evade immune responses and regulate its life cycle. Viral proteins like Vif and Vpu target host immune factors for degradation via the ubiquitin–proteasome pathway. Vif degrades APOBEC3G, ensuring viral replication, while Vpu downregulates CD4 to enhance viral assembly [97]. Nef manipulates ubiquitination to evade immune detection by degrading signaling proteins and MHC-I molecules [98,99].

Autophagy regulates necroptosis. Autophagy can suppress necroptosis by selectively degrading key mediators of the necroptotic pathway, such as RIPK1 and RIPK3, through autophagic degradation. This occurs when ubiquitinated RIPK1 and RIPK3 are recognized by p62/SQSTM1, an autophagy adaptor protein, and directed into autophagosomes for lysosomal degradation [26,100].

By reducing the availability of RIPK1 and RIPK3, autophagy prevents the assembly of the necrosome complex and subsequent activation of MLKL. In HIV-1 infection, the viral manipulation of autophagy disrupts this regulatory mechanism, potentially leading to uncontrolled necroptosis and heightened immune activation [100]. Conversely, necroptosis can influence autophagy by altering cellular stress responses and signaling pathways. For example, DAMPs released during necroptosis may activate autophagy as a compensatory mechanism to mitigate inflammation. However, HIV-1’s manipulation of autophagy may limit this protective effect, exacerbating disease progression [101].

Autophagy influences ferroptosis, which is triggered by an imbalance in iron homeostasis, glutathione metabolism, and lipid peroxidation, making it a critical determinant in the survival and function of immune cells during infection. The following key mechanisms drive this form of cell death: iron dysregulation, glutathione peroxidase 4 (GPX4) inhibition, lipid peroxidation and membrane damage, and cysteine and glutathione deficiency. 

Autophagy influences ferroptosis through the selective degradation of ferritin, a process known as ferritinophagy. Ferritin, the primary intracellular iron storage protein, sequesters excess iron to prevent its participation in harmful Fenton reactions that generate reactive oxygen species [ROS]. During ferritinophagy, the autophagy adaptor protein NCOA4 binds to ferritin and targets it to the autophagosome, where it is subsequently degraded in the lysosome. This degradation releases free iron into the cytoplasm, increasing the labile iron pool. Elevated iron levels catalyze the production of lipid peroxides through Fenton chemistry and the activity of lipoxygenases, driving ferroptosis [102,103].

In the context of HIV-1 infection, dysregulated autophagy can enhance ferritinophagy, leading to increased oxidative stress and lipid peroxidation. Viral proteins such as Tat and Vpr have been shown to enhance mitochondrial ROS production, further promoting lipid peroxidation and ferroptotic cell death. This exacerbates cell death and tissue damage, particularly in immune cells and mucosal tissues critical for controlling HIV-1 progression [104]. The autophagy adaptor protein NCOA4 mediates ferritinophagy, linking autophagy to iron metabolism. In HIV-1 infection, dysregulated autophagy may enhance ferroptosis by elevating free iron and oxidative stress. Ferroptosis can impact autophagy through the generation of lipid peroxidation products, such as malondialdehyde [MDA] and 4-hydroxynonenal [4-HNE], which act as signaling molecules to activate autophagy [105]. These reactive aldehydes can modify key autophagic proteins or signaling pathways, leading to the induction of autophagic responses. For example, lipid peroxidation can activate AMPK, a master regulator of autophagy, by altering cellular energy states or directly modulating signaling pathways [106]. This oxidative burden may also contribute to the persistence of viral reservoirs by selectively eliminating uninfected immune cells while allowing infected cells to survive. In addition to its impact on immune cells, ferroptosis has been implicated in the progression of HIV-1-associated neurocognitive disorders. HIV-1-infected astrocytes and microglia exhibit increased iron accumulation and oxidative damage, leading to neuronal dysfunction and cognitive decline. Understanding how ferroptosis contributes to HAND could pave the way for novel neuroprotective therapies in HIV-1-positive individuals [104].

Autophagy suppresses pyroptosis through several mechanisms, primarily by degrading inflammasome components, such as NLRP3, and mitigating mitochondrial ROS production. NLRP3, a key component of the inflammasome, is selectively degraded via autophagy [mitophagy] when mitochondria are damaged, preventing excessive inflammasome activation and downstream pyroptosis. Mitochondrial ROS serve as important signaling molecules for inflammasome activation; therefore, the removal of dysfunctional mitochondria through mitophagy reduces ROS levels and inflammasome activity [107].

In the context of HIV-1 infection, viral proteins such as Tat and Nef disrupt autophagy by inhibiting autophagosome maturation or promoting mitochondrial damage. This disruption leads to an accumulation of damaged mitochondria and elevated ROS levels, enhancing inflammasome activation and pyroptosis. Additionally, HIV-1-induced metabolic stress exacerbates mitochondrial dysfunction, further impairing autophagy and creating a positive feedback loop that amplifies inflammation and immune cell death [32].

Understanding these mechanisms is critical for identifying therapeutic targets to restore balance between autophagy and pyroptosis in HIV-1 pathogenesis.

## 5. Autophagy and Cell Death Mechanisms in Clearing HIV Reservoirs: Novel Therapeutic Strategies

HIV-1 persists in patients despite antiretroviral therapy (ART) (Appendix A: Summary of HIV Therapeutic Drugs and Their Targeted Pathways) due to the presence of latent viral reservoirs, primarily in resting CD4+ T cells and other immune cell populations. These reservoirs present a significant challenge to achieving a cure for HIV-1 [108]. Recent studies have focused on leveraging cellular processes such as autophagy and regulated cell death mechanisms to target and eliminate these reservoirs. Autophagy, a cellular degradation pathway, and various forms of programmed cell death, including apoptosis, necroptosis, ferroptosis, and pyroptosis, offer promising strategies for reducing or eradicating latent HIV-1 reservoirs [26].

As previously mentioned, autophagy plays a dual role in HIV-1 infection, acting both as a host defense mechanism and as a pathway manipulated by the virus to promote its survival. These mechanisms can be harnessed as therapeutic strategies to counteract viral persistence. One approach is to induce autophagy to degrade latent HIV-1 components; experimental models demonstrate that enhancing autophagic flux can lead to the degradation of viral proteins and RNA, reducing the reservoir size. Autophagy inducers such as rapamycin and metformin have shown promise in activating autophagy and targeting latent reservoirs [26]. Another type of approach described in a recent work regards the induction of a specific form of autophagy-dependent cell death known as autosis. Autosis is characterized by unique morphological and biochemical features and is regulated by the Na+/K+-ATPase enzyme. The researchers utilized nanopeptides to trigger autosis selectively in latently HIV-infected cells, aiming to reduce the viral reservoir that persists despite antiretroviral therapy. This strategy holds potential for targeting and eliminating cells harboring latent HIV, contributing to efforts toward a functional cure for HIV infection [109].

A second approach is to regulate autophagy and immune surveillance. Autophagy can enhance the presentation of viral antigens on major histocompatibility complex (MHC) molecules, thereby promoting the immune recognition of HIV-1-infected cells by cytotoxic T lymphocytes (CTLs). This pathway could be exploited to “unmask” latent reservoirs for immune-mediated clearance [32]. Additionally, autophagy dysregulation in chronic HIV-1 infection is associated with persistent immune activation, which contributes to the progression of non-AIDS-related comorbidities, such as cardiovascular and neurocognitive disorders [110]. Modulating autophagy could, therefore, offer broader therapeutic benefits beyond reservoir clearance.

Programmed cell death pathways offer additional strategies to eliminate HIV-1 reservoirs by selectively targeting infected cells. Latency reversal agents (LRAs), which reactivate latent viruses, make infected cells more susceptible to apoptosis. Combination therapies using LRAs and pro-apoptotic agents, such as BH3 mimetics, have been effective in inducing cell death in reactivated reservoirs [111]. Necroptosis, a regulated form of necrosis mediated by RIPK1 and RIPK3 kinases, results in the release of damage-associated molecular patterns (DAMPs) and immunogenic cell death. Targeting HIV-1-infected cells via necroptosis, by modulating the RIPK1-RIPK3-MLKL axis, has emerged as a promising strategy for reservoir elimination [71].

Targeting ferroptosis in HIV-1 reservoirs may selectively kill infected cells due to their increased metabolic activity and oxidative stress. Ferroptosis inducers, such as erastin, have been proposed as potential agents for reservoir clearance [32]. Pyroptosis, a pro-inflammatory form of cell death driven by caspase-1 activation and gasdermin D pore formation, is a major driver of CD4+ T cell depletion during HIV-1 infection. Strategies to induce pyroptosis specifically in reservoir cells could enhance infected cell clearance while minimizing systemic inflammation [76]. Given that direct cell-to-cell transmission of HIV-1 is essential for inducing pyroptosis, targeting the virological synapse could represent a novel therapeutic strategy to prevent CD4+ T cell depletion and slow HIV-1 disease progression [110].

To maximize the elimination of HIV-1 reservoirs, combining autophagy modulation with cell death pathways holds significant promise. LRAs can reactivate latent HIV-1, making the virus and its host cells susceptible to autophagic degradation or immune clearance. Combining autophagy modulation with cell death pathways holds significant promise for eliminating HIV-1 reservoirs. Latency reversal agents (LRAs) can reactivate latent HIV-1, rendering the virus and its host cells susceptible to autophagic degradation or immune clearance. For instance, the combination of the LRA ingenol dibenzoate (IDB) with the BCL-2/BCL-xL inhibitor ABT-263 and autophagy inhibitors has been shown to induce substantial cell death in HIV-infected T cells, achieving over 90% elimination of HIV-1-p24+ T cells in vitro within two days [111]. Additionally, since HIV-1-infected cells exhibit altered metabolic states, such as increased reactive oxygen species (ROS) and lipid peroxidation, exploiting these vulnerabilities through ferroptosis inducers and autophagy modulators could provide a novel therapeutic avenue.

HIV-1 manipulates autophagy and programmed cell death pathways to ensure its survival and replication. By promoting Nef-mediated BCL2 ubiquitination at the endoplasmic reticulum (ER), HIV-1 blocks autophagy, aiding viral persistence. Conversely, the enhancement of PRKN-mediated BCL2 ubiquitination at mitochondria induces apoptosis, contributing to immune cell destruction and HIV-1 progression. The depletion of CD4+ T cells, caused by direct infection and bystander apoptosis, leads to immunosuppression, increasing vulnerability to opportunistic infections and malignancies [37]. Chronic activation of apoptotic pathways and autophagy disruption results in ongoing immune activation, even in ART-treated patients, further linking these pathways to non-AIDS-related comorbidities [76].

Targeting the ubiquitination process represents a potential therapeutic strategy in HIV-1 infection. Furthermore, modulating the interplay between autophagy, pyroptosis, and necroptosis offers additional opportunities. Pharmacological agents such as rapamycin or chloroquine can enhance autophagy’s anti-inflammatory and antiviral effects [37]. Inhibitors of necroptosis, such as RIPK1 inhibitors, may reduce immune activation and tissue damage, potentially improving clinical outcomes in HIV-1 patients. Conversely, inhibitors of pyroptosis, such as caspase-1 inhibitors, could mitigate CD4+ T cell loss and chronic inflammation. A combined approach targeting multiple cell death pathways could improve immune function and reduce HIV-1-associated tissue damage [32].

Finally, ferroptosis-targeted therapies could be explored further in HIV-1 pathogenesis. Since ferroptosis is closely linked to oxidative stress and lipid peroxidation, therapeutic strategies could aim to mitigate these processes to limit HIV-1 reservoir survival. Deferoxamine and other iron chelators may reduce ferroptotic cell death by limiting iron availability as it was demonstrated in musculoskeletal diseases by Zhang et al. [112]. Targeting GPX4 with selenium supplementation or pharmacological activators may counteract lipid peroxidation and prevent ferroptosis [113]. N-acetylcysteine (NAC) and vitamin E can enhance glutathione levels and protect against oxidative damage [114].

## 6. Discussion

The review delves into the dynamic interplay between autophagy and programmed cell death modalities within the context of HIV-1 pathogenesis, uncovering the multifaceted roles these processes play in disease progression and viral persistence. Both autophagy, a conserved homeostatic process, and programmed cell death modalities serve as essential regulators of HIV-1 replication and immune response, offering potential targets for therapeutic interventions.

A key insight from this review is that autophagy exhibits dual roles in HIV-1 infection. On one hand, autophagy serves as a protective host mechanism by degrading viral components and reducing HIV-1 replication. On the other hand, HIV-1 manipulates autophagic pathways to promote viral survival, using proteins like Nef to inhibit autophagosome formation and disrupt antigen presentation, thereby evading immune detection. The virus further exploits autophagy by inducing selective mitophagy, via PINK1, to maintain mitochondrial function and provide an environment conducive to replication. This nuanced relationship between HIV-1 and autophagy underscores the complexity of autophagic manipulation and highlights the challenges in developing therapies that can selectively promote antiviral autophagy without enhancing viral replication.

Apoptosis, the most relevant programmed cell death mechanism involved in HIV-1 pathogenesis, conversely, presents a more straightforward, albeit devastating, role in HIV-1 pathogenesis. The induction of apoptosis in CD4+ T cells, whether directly by viral proteins like Vpr, Tat, and Nef, or indirectly through bystander effects and inflammatory responses, serves as a major contributor to immune dysfunction and AIDS progression. The ability of HIV-1 to induce apoptosis in uninfected bystander cells further exacerbates CD4+ T cell depletion, compromising immune defense. Notably, this review sheds light on the role of HIV-1 proteins in modulating apoptotic pathways, with the activation of caspases, disruption of mitochondrial integrity, and dysregulation of Bcl-2 family proteins contributing to both viral persistence and immune collapse.

The discussion on the crosstalk between autophagy and apoptosis reveals an intricate regulatory network where both processes converge through shared molecular regulators such as BECN1, p53, and caspases. This cross-regulation implies that the balance between autophagy and apoptosis is crucial in determining the fate of HIV-1-infected cells. HIV-1 manipulates this balance, inhibiting autophagy to avoid degradation while promoting apoptosis to eliminate immune cells and sustain viral reservoirs. Additionally, the role of ubiquitination, particularly through Parkin and BCL2, highlights the complex regulation of these pathways in the viral life cycle, offering another layer of potential therapeutic intervention.

This review demonstrates that targeting the interplay between autophagy and apoptosis may provide promising avenues for HIV-1 therapy. However, therapeutic approaches must be highly specific, as improper modulation of these processes could either enhance viral replication or cause unintended damage to immune cells. Future research should prioritize unraveling the specific molecular interactions that govern autophagy and apoptosis during HIV-1 infection and explore how these insights can be harnessed for targeted therapies aimed at eradicating viral reservoirs and preventing immune system deterioration.

## 7. Conclusions

The interplay between autophagy and programmed cell death modalities in HIV-1 pathogenesis offers a profound insight into the virus’s ability to exploit host cellular mechanisms for its survival and propagation. While autophagy serves as a double-edged sword, acting both as a host defense and as a pathway manipulated by HIV-1 to ensure its persistence, programmed cell death mechanisms like apoptosis, necroptosis, pyroptosis, and ferroptosis further illustrate the intricate dynamics of immune evasion, viral replication, and immune cell depletion.

HIV-1’s ability to manipulate these processes underscores the complexity of its pathogenesis, with viral proteins playing pivotal roles in disrupting cellular homeostasis and immune function. The virus strategically balances autophagic suppression and activation, apoptosis induction, and bystander cell death to maximize its replication while minimizing immune clearance. This review highlights the multifaceted roles of these pathways, particularly the convergence of autophagy and apoptosis, and their regulatory mechanisms, including ubiquitination and mitochondrial quality control.

Targeting these cellular processes holds significant potential for advancing HIV-1 therapeutics. Strategies aimed at restoring the antiviral functions of autophagy, inhibiting excessive apoptosis, and selectively inducing other cell death modalities in viral reservoirs could revolutionize current approaches to HIV-1 management. However, these interventions require precision to avoid unintended consequences, such as exacerbating immune dysfunction or promoting viral persistence.

In conclusion, a deeper understanding of the molecular underpinnings of autophagy and programmed cell death in the context of HIV-1 infection is critical for developing innovative therapeutic strategies. Future research should focus on elucidating these intricate host–virus interactions and translating these findings into clinical interventions to achieve the ultimate goal of viral eradication and immune restoration.

## Data Availability

No new data were created or analyzed in this study. Data sharing is not applicable to this article.

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
