# Peer review of "Autophagy and Programmed Cell Death Modalities Interplay in HIV Pathogenesis"

_cells, 2025, doi:10.3390/cells14050351_

Round 1

Reviewer 1 Report

Comments and Suggestions for Authors

This is an interesting review dealing with the processes of cell death and the context of HIV-1 infection. It is well written, well-conceived, and open new perspective on manipulating cell death processes to eliminate HIV-1 reservoir.

Some minor points need to be addressed however:

1-    HIV-1 should replace HIV all over the review. HIV-1 pathogenesis is different from HIV-2

2-    The sentence on page 4 line 164, 165 “In addition, PINK1-regulated mitophagy

can influence the presentation of viral antigens and the activation of adaptive immune cell

response” lacks references.

3-    The paragraph on page 4 from line 167 to 177 regarding sting needs references.

The font of references 49 to 51 on page 5 needs to match the text font.

4-    PML is mentioned on page 6 line 207. Define PML

5-    Font of reference 56 on page 6 line 215 needs to be fixed

6-    These sentences on page 6 from line 219 to 224 lack of references. “Apoptosis in CD4+ T cells during HIV Infection can be induced through the action of several viral proteins like Vpr, which disrupt mitochondrial membrane potential, leading to the activation of caspases and cell death (REF). Vif also has an indirect role in inducing apoptosis in uninfected bystander CD4+ T cells through the release of viral proteins and inflammatory cytokines, contributing to overall CD4+ T cell Depletion” (REF).

7-    Define RIPK1, RIPK3, and MLKL on page 6 line 250. What are these proteins?

8-    This sentence of page 7 line 282-283 needs references “Elevated levels of iron and ROS, along with impaired antioxidant defenses, have been observed in HIV-infected cells” (REF).

9-    The paragraph on page 11 line 423, to 426 “HIV persists in patients despite antiretroviral therapy (ART) due to the presence of latent viral reservoirs, primarily in resting CD4+ T cells and other immune cell populations. These reservoirs present a significant challenge to achieving a cure for HIV. Recent studies have focused on leveraging cellular processes such as autophagy and regulated cell death mechanisms to target and eliminate these reservoirs” lacks references.

10- Reference 109 to which to author refer to on page 12 line 443 is out of context, as reference 109 is a review not related to HIV-1 reservoir and on the mechanisms to unmask HIV-1 reservoir.

Author Response

Dear Reviewer,

Thank you for your thoughtful and positive feedback on our review. We appreciate your recognition of the manuscript’s contribution to understanding cell death processes in the context of HIV-1 infection and their potential implications for targeting the viral reservoir.

Regarding your specific comments:

  1. HIV-1 should replace HIV throughout the review. HIV-1 pathogenesis is different from HIV-2.
    We acknowledge the distinction between HIV-1 and HIV-2 pathogenesis and appreciate your suggestion. We have carefully reviewed the manuscript and replaced all instances of "HIV" with "HIV-1" to ensure accuracy and clarity.
  2. The sentence on page 4, lines 164–165: “In addition, PINK1-regulated mitophagy can influence the presentation of viral antigens and the activation of adaptive immune cell response” lacks references.
    Lines 164-165 are now lines 234-236

We acknowledge the need for proper citation to support this statement. We have now incorporated the following reference to substantiate this claim:

[46] Diana Matheoud, Tyler Cannon, Aurore Voisin, Anna-Maija Penttinen, Lauriane Ramet, Ahmed Fahmy, Charles Ducrot, Annie Laplante, Marie-Josée Bourque, Lei Zhu, Armelle Le Campion, Heidi McBride, Samantha Gruenheid, Louis-Eric Trudeau, Michel Desjardins; Parkinson’s disease related proteins PINK1 and Parkin are major regulators of the immune system. J Immunol 1 May 2019; 202 (1_Supplement): 177.27. https://doi.org/10.4049/jimmunol.202.Supp.177.27

  1. The paragraph on page 4, lines 167–177 regarding STING needs references.
    Lines167-177 are now 238-248

 We recognize the need for proper referencing and have now added the following references to support the statements made in this section:

[51]Schmid, M., Fischer, P., Engl, M., Widder, J., Kerschbaum-Gruber, S., & Slade, D. (2024). The interplay between autophagy and cGAS-STING signaling and its implications for cancer. Frontiers in immunology15, 1356369. https://doi.org/10.3389/fimmu.2024.1356369

[52]Sumner, R. P., Harrison, L., Touizer, E., Peacock, T. P., Spencer, M., Zuliani-Alvarez, L., & Towers, G. J. (2020). Disrupting HIV-1 capsid formation causes cGAS sensing of viral DNA. The EMBO journal39(20), e103958. https://doi.org/10.15252/embj.2019103958

    •  
  1. The font of references 49 to 51 on page 5 needs to match the text font.
    We appreciate your attention to formatting consistency. We have corrected the font of references 49 to 51 to ensure uniformity.
  2. PML is mentioned on page 6, line 207. Define PML.
    Line 207 is now 306
  3. We have now provided a definition for PML (Promyelocytic Leukemia Protein) at its first mention for clarity:
    • PML is a key component of PML nuclear bodies (PML-NBs), which play a role in cellular processes such as DNA damage response and antiviral defense mechanisms.
  1. [57] Perfettini JL, Nardacci R, Séror C, Bourouba M, Subra F, Gros L, Manic G, Amendola A, Masdehors P, Rosselli F, Ojcius DM, Auclair C, de Thé H, Gougeon ML, Piacentini M, Kroemer G. The tumor suppressor protein PML controls apoptosis induced by the HIV-1 envelope. Cell Death Differ. 2009 Feb;16(2):298-311. doi: 10.1038/cdd.2008.158. Epub 2008 Nov 21. PMID: 19023333.
  1. The font of reference 56 on page 6, line 215 needs to be fixed.
    We have corrected the font of reference 56 to ensure consistency with the rest of the text.
  2. The sentences on page 6, lines 219–224 lack references.
    Lines 219-224 are now 319-324 We have now added the following references to support these statements:

[59] Muthumani, K., Hwang, D.S., Desai, B.M., Zhang, D., Dayes, N., Green, D.R. and Weiner, D.B. (2002). HIV-1 Vpr Induces Apoptosis through Caspase 9 in T Cells and Peripheral Blood Mononuclear Cells. Journal of Biological Chemistry, 277(40), pp.37820–37831. doi:https://doi.org/10.1074/jbc.m205313200.

[60] Mbita Z, Hull R, Dlamini Z. Human immunodeficiency virus-1 (HIV-1)-mediated apoptosis: new therapeutic targets. Viruses. 2014 Aug 19;6(8):3181-227. doi: 10.3390/v6083181. PMID: 25196285; PMCID: PMC4147692. Badley, A. D

[61 L. Conti, G. Rainaldi, P. Matarrese, B. Varano, R. Rivabene, S. Columba, A. Sato, F. Belardelli, W. Malorni, S. Gessani; The HIV-1 vpr Protein Acts as a Negative Regulator of Apoptosis in a Human

  1. Define RIPK1, RIPK3, and MLKL on page 6, line 250.
    Line 250 is now 388
  2. We have now provided clear definitions for these proteins:
    • RIPK1 (Receptor-Interacting Protein Kinase 1): RIPK1 is a crucial mediator in necroptosis, apoptosis, and inflammation. Upon activation by death receptors like TNFR1, RIPK1 can interact with RIPK3 to form a complex known as the necrosome, initiating necroptosis. In HIV-infected cells, the viral envelope glycoprotein gp41 can induce necroptosis through a pathway involving RIPK1, contributing to the depletion of CD4+ T cells. 
    • RIPK3 (Receptor-Interacting Protein Kinase 3): RIPK3 is essential for the execution of necroptosis. It interacts with RIPK1 to form the necrosome, leading to the phosphorylation and activation of MLKL. In the context of HIV, RIPK3-mediated necroptosis has been implicated in the death of infected cells, as well as uninfected bystander CD4+ T cells, thereby contributing to immune system decline. 
    • MLKL (Mixed Lineage Kinase Domain-Like Protein):  MLKL is the downstream effector in the necroptosis pathway. Once phosphorylated by RIPK3, MLKL translocates to the plasma membrane, causing its permeabilization and leading to cell death. In HIV infection, the activation of MLKL by RIPK3 is a critical step in the necroptosis of CD4+ T cells, contributing to the immunodeficiency characteristic of AIDS. 
  3. The sentence on page 7, lines 282–283 “Elevated levels of iron and ROS, along with impaired antioxidant defenses, have been observed in HIV-infected cells” needs references.
    We have expanded the whole section answering to the request of another reviewer so the previously mentioned sentence in not now found.
  4. The paragraph on page 11, lines 423–426 lacks references.

The lines 423–426 are now lines 649-652
We have now incorporated relevant citation n 101:

[101]Campbell, G. R., & Spector, S. A. (2021). Induction of Autophagy to Achieve a Human Immunodeficiency Virus Type 1 Cure. Cells10(7), 1798. https://doi.org/10.3390/cells10071798

  1. Reference 109 on page 12, line 443 is out of context.

Line 443 is now line 808

 We acknowledge this inconsistency and have now replaced reference 109 with a more appropriate citation:

[38] Klute, S., & Sparrer, K. M. J. (2024). Friends and Foes: The Ambivalent Role of Autophagy in HIV-1 Infection. Viruses16(4), 500. https://doi.org/10.3390/v16040500Chen, G., Liu, Y., Wang, Y., Wang, Q., Huo, S., Zhang, X., Cao, Z., Song, M. and Li, Y. (Year). PINK1/Parkin-mediated mitophagy is activated to protect against AFB1-induced immunosuppression in mice spleen. Toxicology Letters. [Online]. Available at: https://www.liankebio.com/citations/28750

We appreciate your meticulous review, which has helped us improve the accuracy and relevance of our manuscript. Please let us know if you have any further recommendations.

Best regards.

Reviewer 2 Report

Comments and Suggestions for Authors

The review entitled Autophagy and programmed cell death modalities interplay in HIV pathogenesis aims to decipher how deregulated autophagy in people living with HIV-1 contributes to cell death and impacts HIV reservoir. Although mainly focusing on HIV-1 productive infection along with the impact of HIV regulatory proteins on autophagy, the authors still provide some insight onto the fact that strong autophagy is also a signature of naturally protected elite controllers (ECs) in the context of HIV aviremia.

The research topic of this review is timely, and the figures are well structured.

A)      Minor concerns

. The tables and figures must be included in the text once they are referred; For instance, figure 1 appears at the end of the review, and required to find it. Also, Table 3 is referred in the text (line 199), before Table 2 (line 226). This must be introduced in order.

. Although mainly focusing on how HIV regulatory proteins impair autophagy and several cell deaths, the review still emphasizes that "In particular, autophagy was shown to be effective in controlling HIV-1 infection among long-term non-progressors and ECs. These individuals maintain stable health without antiretroviral therapy, suggesting unique mechanisms at play." (Lines 80-83).

Those mechanisms are now being identified and addressed in few publications. For example, a. Angin M. et al, Nature Metabolism, 2019; DOI: 10.1038/s42255-019-0081-4: The data show that patients under ART display excessive use of the glycolysis, when ECs are depending on mitochondrial input; b. and c. Loucif H, et al. Autophagy; 2022 and 2021; DOI: 10.1080/15548627.2021.1972403 and DOI: 10.1080/15548627.2021.1874134: The data show that strong autophagy in both HIV-specific cd4 and cd8 T-cells of ECs is key for ensuring optimal immune protection; and d. and e. : Mu W. et al, JCI Insight, 2022; DOI: 10.1172/S0007114520002858: Data show that autophagy inducer rapamycin treatment of HIV-infected humanized mice results in improved antiviral T-cell functions, lower viral loads and PTC after ART cessation.

. One critical publication is missing on the section 5. Autophagy and cell death mechanisms in clearing HIV reservoirs (line 422). Zhang G, Cell death Dis., 2019; DOI: 10.1038/s41419-019-1661-7.

. Although the authors are using a significant number of reviews instead of focusing on the specific published studies. I would advice the authors to find the studies highlighting their points.

B)      Major concerns

The major concern of the review (after doing the exercise of verifying my point) is that many information that are provided in the text are not consistent with their references and are even extrapolated by using distinct study models.

For example, lines 240 to 249: The authors provide several references which they used to say that mitophagy protein PINK1 influences apoptosis in the context of HIV-1 infection; It is not accurate and is only an interpreted extrapolation by suing different studies and reviews. Reference 65 is a review showing the role of PINK1 on longevity and not in HIV context; reference 66 is addressing PINK1 on neurodegenerative diseases; reference 67 is addressing the role of mitochondrial OxPHOS and ROS production in HIV; reference 68  is showing that PINK1 inhibits osteoclast apoptosis and once again not on HIV context, and reference 69 is a review addressing the interplay between mitophagy and apoptosis.

Another example is shown in the section autophagy regulates necroptosis (Lines 340-344); reference 102, once again is a review and is addressing of the ubiquitination of BCL2 and impact on autophagy in HIV context and does not address both RIPK1 and RIPK3; reference 103: impact of HIV proteins on proteasomal process and not viral manipulation of autophagy, etc.

In the same vein, lines 316 to 317; reference 82 is showing that elevated IL-1b, even under ART, impairs the IL-7 responsiveness in memory CD4 T-cells, and not addressing any link with autophagy BECN1 as stipulated by the authors; reference 83 is on colon cancer and not HIV context; and reference 84 is also on cancer model (melanoma).

To summarize, I would advise the authors to reverify every reference, and the information that are provided in the text to be consistent with them; a review must summarize precisely and give an update with the most recent study publications, instead of providing extrapolated information. However, after revising and being ensured of all study data on autophagy and cell death in PLWHs, the review could be a good addition.

Author Response

Dear Reviewer,

Thank you for your valuable feedback. We sincerely appreciate your careful review of our manuscript and your insightful comments, which have helped us refine and improve its quality.

Revisions to Tables and Figures Placement

We have revised the document to ensure that all tables and figures are placed immediately after they are first mentioned in the text. Specifically:

  • The order of Tables 2 and 3 has been corrected to ensure they appear sequentially as referred to in the text.
  • Figure 1 has been repositioned in the supplementary materials  to prevent any difficulty in locating it.

Integration of Key Studies

We appreciate your suggestion to incorporate additional key studies to enhance our discussion. We have now included:

[16] The findings from Angin et al. (2019) found at line 86 , highlighting metabolic differences between ECs and ART-treated individuals.

These additions strengthen our discussion on how autophagy contributes to immune protection and viral control in ECs. We appreciate your guidance in refining this aspect of our review.

Inclusion of Missing Critical Publication

Thank you for recommending the study by Zhang et al. (2019) included as reference n [112] found at line 807 in Section 5 on autophagy and cell death mechanisms in clearing HIV reservoirs. We have now incorporated this reference and revised the discussion accordingly. The findings from Zhang et al. (2019) provide important insights into the interplay between autophagy and cell death in the context of HIV reservoir clearance, further enriching our analysis.

Prioritization of Primary Research Studies

We acknowledge your suggestion to rely more on original research studies rather than reviews to support our key points. In response, we have:

Revisited our references and replaced or supplemented some review articles with primary research studies that provide direct experimental evidence for the mechanisms discussed.

We appreciate your constructive advice, which has helped improve the quality of our review.

Verification of References and Accuracy of Claims

Thank you for your thorough review and for highlighting inconsistencies between some references and the information provided in the manuscript. In response, we have conducted a comprehensive verification of all references. Specifically, we have:

  • Reassessed the references in lines 240–249 now lines 377-387 regarding the role of PINK1 in apoptosis in the context of HIV-1 infection. We acknowledge that some of the previous references did not directly support this relationship. We have now revised this section
  • Corrected the section on autophagy regulation of necroptosis (Lines 340–344) by replacing or modifying references that did not directly address RIPK1 and RIPK3 found at line 567in the context of HIV.
  • Reviewed the discussion on IL-1β and IL-7 responsiveness (Lines 316–317) now lines 484-485 and replaced references that were not directly related to autophagy in HIV. We recognize that some of the previous references focused on cancer models, which may not be directly applicable to our review.
  • Conducted a broader review of all references throughout the manuscript to ensure that every claim made in the text is accurately supported by appropriate and primary research studies.

Final Remarks

We are grateful for your careful review and guidance in improving the scientific accuracy and credibility of our manuscript. We believe these revisions have significantly strengthened the quality and reliability of our work.

Best regards.

Reviewer 3 Report

Comments and Suggestions for Authors

This review extensively discusses the roles of autophagy and several programmed cell death pathways (apoptosis, necroptosis, ferroptosis, pyroptosis) in HIV pathogenesis.  They summarized detailed signaling responsible for the interactions between autophagy and the other death pathways in the context of chronic HIV infection.  Overall, the review summarizes the current acknowledges/understanding on this important topic and gives us a holistic view.  Still, the reviewer has some suggestions:

1) Overall, the description/writing is not in concise.  For example, there are some repetitive sentences on autophagy and apoptosis in different parts.  The writing needs more edits. 

2) The therapy part is not in well organized.  The section 5 and 6 can be merged into one with more information. The authors can summarize current drugs are now be used for HIV treatment or eliminating HIV reservoirs and also which pathways they can targets (summarized in one table).  

3) Need a little more discussion on ferroptosis.  The understanding on this specific program cell death pathway progresses rapidly in the last three years.    

Author Response

Dear Reviewer,
We sincerely appreciate your thoughtful review and valuable feedback on our manuscript. Your comments have helped us refine our work to improve clarity, conciseness, and organization. Below, we address each of your suggestions:

1) Overall, the description/writing is not concise. There are repetitive sentences on autophagy and apoptosis in different parts. The writing needs more edits.
•    We have carefully revised the manuscript to improve conciseness, eliminating redundant sentences and streamlining discussions on autophagy and apoptosis to avoid repetition.
•    A thorough language and structural edit were conducted to enhance clarity and readability.

2) The therapy section is not well organized. Sections 5 and 6 should be merged into one with more information. The authors should summarize current drugs used for HIV treatment or eliminating HIV reservoirs and indicate which pathways they target (summarized in one table).
•    We sincerely appreciate your valuable suggestion regarding the organization of the therapy section. Based on your feedback, we have merged Sections 5 and 6 found at line 644, to enhance clarity and coherence. Additionally, we have elaborated on the discussion by summarizing the current drugs used for HIV treatment and strategies aimed at eliminating HIV reservoirs.
•    To further improve the presentation of this information, we have created a comprehensive table that outlines these drugs along with their targeted pathways, as per your recommendation, it will be found in the supplementary material under the name Supplementary Table 1.  We believe this modification strengthens the section and provides a clearer overview of therapeutic approaches.

Table: Summary of HIV Therapeutic Drugs and Their Targeted Pathways
Drug Class    Example Drugs    Targeted Pathway
Nucleoside Reverse Transcriptase Inhibitors (NRTIs)    Zidovudine, Tenofovir    Inhibit reverse transcriptase, preventing viral replication
Non-Nucleoside Reverse Transcriptase Inhibitors (NNRTIs)    Efavirenz, Nevirapine    Bind to reverse transcriptase, causing enzyme dysfunction
Protease Inhibitors (PIs)    Ritonavir, Darunavir    Inhibit HIV protease, preventing viral maturation
Integrase Strand Transfer Inhibitors (INSTIs)    Raltegravir, Dolutegravir    Block viral DNA integration into the host genome
Entry Inhibitors    Maraviroc, Enfuvirtide    Block viral entry by targeting CCR5 or fusion proteins
Immune Modulators    Broadly Neutralizing Antibodies (bNAbs)    Enhance immune response against HIV
Latency-Reversing Agents (LRAs)    Romidepsin, Vorinostat    Reactivate latent HIV reservoirs for immune clearance
Ferroptosis Modulators (Experimental)    Iron Chelators, GPX4 Activators    Prevent ferroptosis-related immune cell loss

3) Need a more detailed discussion on ferroptosis. The understanding of this specific programmed cell death pathway has progressed rapidly in the last three years.
•    We have incorporated additional discussion on ferroptosis found at line 576, emphasizing recent advancements over the past three years.
•    We have highlighted its relevance in HIV pathogenesis and its interactions with autophagy and other cell death pathways, integrating findings from the latest literature.

We believe these revisions have strengthened our manuscript and aligned it with the reviewer's recommendations. We appreciate your valuable time and thoughtful feedback.
Best regards

Round 2

Reviewer 2 Report

Comments and Suggestions for Authors

Better updated version of their previous review; the authors took the time to adjust informations provided in the text and related publications...

Reviewer 3 Report

Comments and Suggestions for Authors

The authors have addressed all concerns raised by the reviewer.